# PET Imaging in Bladder Cancer: An Update and Future Direction

**DOI:** 10.3390/ph16040606

**Published:** 2023-04-17

**Authors:** Jules Zhang-Yin, Antoine Girard, Etienne Marchal, Thierry Lebret, Marie Homo Seban, Marine Uhl, Marc Bertaux

**Affiliations:** 1Department of Nuclear Medicine, Clinique Sud Luxembourg, Vivalia, B-6700 Arlon, Belgium; 2Department of Nuclear Medicine, Amiens-Picardy University Hospital, 80054 Amiens, France; 3Department of Urology, Foch Hospital, 92150 Suresnes, France; 4Department of Nuclear Medicine, Foch Hospital, 92150 Suresnes, France; 5Department of Urology and Renal Transplantation, Amiens-Picardy University Hospital, 80054 Amiens, France

**Keywords:** positron emission tomography, bladder cancer, 2-[18F]fluoro-2-deoxy-D-glucose

## Abstract

Molecular imaging with positron emission tomography is a powerful tool in bladder cancer management. In this review, we aim to address the current place of the PET imaging in bladder cancer care and offer perspectives on potential future radiopharmaceutical and technological advancements. A special focus is given to the following: the role of [^18^F] 2-[18F]fluoro-2-deoxy-D-glucose positron emission tomography in the clinical management of bladder cancer patients, especially for staging and follow-up; treatment guided by [^18^F]FDG PET/CT; the role of [^18^F]FDG PET/MRI, the other PET radiopharmaceuticals beyond [^18^F]FDG, such as [^68^Ga]- or [^18^F]-labeled fibroblast activation protein inhibitor; and the application of artificial intelligence.

## 1. Introduction

Bladder cancer (BC) is the 10th most diagnosed cancer globally, with more than 500,000 new cases and around 200,000 deaths in 2018 [1]. The most important risk factor for developing BC is tobacco smoking, followed by occupational exposure to aromatic amines. In developed countries, more than 90% of BCs are urothelial carcinomas (UC), including a conventional subtype and histomorphologic variants (i.e., microcystic, nested, plasmocytoid etc.), some of which have prognosis implications [2]. Squamous cell carcinomas account for 5% of BC in developed countries, but they are more common in areas where Schistosoma haematobium is endemic. Lastly, 2% of BCs are adenocarcinomas [3]. Patient with BC usually present with painless haematuria. The final diagnosis is made by a histological analysis of tissue resected during transurethral resection of the bladder tumor (TURBT). When it shows non-muscle invasive bladder cancer (NMIBC), the disease can be managed with local treatments and tends to recur but is generally not life-threatening. A systemic imaging work-up for metastatic disease is not needed in that case, except for a few patients with high-risk NMIBC. When muscle-invasive bladder cancer (MIBC) is present (around 30% of patients), i.e., ≥pT2 in the WHO classification [4,5], a regional and distal cross-sectional imaging work-up must be performed to search for lymph node and distant metastases, as well as concomitant upper urinary tract urothelial carcinoma. According to current guidelines (ESMO, EAU, AUA), it must include a contrast-enhanced computed tomography (CT) or magnetic resonance imaging (MRI) of the abdomen-pelvis combined with a chest CT [6]. To date, [^18^F] 2-[18F] fluoro-2-deoxy-D-glucose ([^18^F]FDG) positron emission tomography (PET) is not systematically recommended but is increasingly used in clinical practice. The treatment of MIBC is evolving rapidly. Cisplatin-containing combination chemotherapy is the first-line standard in advanced or metastatic patients fit enough to tolerate cisplatin. Cisplatin-ineligible patients with programmed death-ligand 1 (PD-L1) expression or those who progress after first-line chemotherapy can be treated with immune checkpoint inhibitors (ICI). Targeted therapy such as erdafitinib or antibody-drug conjugate enfortumab-vedotin can be given if another progression occurs. In patients with no abdominal or pelvic wall invasion and no lymph node or distant metastatic disease (i.e., T2-T4a, N0 M0 MIBC), the standard treatment is radical cystectomy (RC) with extended pelvic lymph node dissection (ePLND). The prognostic value of the ePLND is important, as the 5-year survival is only 26% when it shows lymph node metastasis [4]. Cisplatin-based neoadjuvant chemotherapy (NAC) is recommended before surgery because it was associated with an absolute increase of 5% in 5-year overall survival in a meta-analysis of 11 randomized trials of 3005 patients [7]. However, the value of NAC is questioned by some who believe that it may only benefit those with the most advanced disease. In some patients, a tri-modality combination of TURBT, radiotherapy, and chemotherapy can be offered as an alternative to RC.s. Cisplatin-based preoperative chemotherapy with cross-sectional imaging evaluation can be offered in some patients with only pelvic lymph node metastases (i.e., N1 or N2), followed by RC and ePLND when they do not experience progression during treatment. Lastly, ICI is currently studied as an adjuvant treatment in patients who have undergone surgery, with promising results in terms of disease-free survival [2]. Thus, cross-sectional imaging is a cornerstone in the management of patients with MIBC at initial staging particularly, but also during follow-up. PET imaging provides quantitative images of biological processes. It is systematically coupled with a CT (PET/CT), or less often with an MRI (PET/MRI). [^18^F]FDG-PET/CT hass increasingly been used over the years in clinical practice, but the low evidence level of published studies hampers its systematic use being implemented in international guidelines. However, the data are progressively accumulating, and the technique is evolving, with better quality images (better spatial resolution and higher sensitivity) and the development of new radiopharmaceuticals. The purpose of this article is to provide an up-to-date review of the performance and potential uses of [^18^F]FDG-PET in the management of bladder cancer, and to discuss possible future developments in the field. We have searched in the database of MEDLINE using the key words of “PET” and “bladder cancer”. The literature essentially covers papers published after 2010. All authors participated in selecting the papers, thanks to their experience in the field.

## 2. [^18^F]FDG PET/CT

### 2.1. Initial Staging and Relapse

#### 2.1.1. Primary Tumor Evaluation

The evaluation of primary tumor in the bladder is mainly performed during cystoscopy. Superficial lesions are directly visible in most cases, and TURBT sample analysis is used to search for bladder muscle invasion. However, TURBT staging performances can vary among urologists. An upstaging from NMBIC to MIBC can happen in 32% of patients when a second TURBT or cystectomy is performed [8]. MRI of the bladder can help to specify the depth of neoplastic infiltration in patients. With a dedicated scoring system called VIRADS [9], it can accurately help to differentiate MIBC from NMIBC (sensitivity of 90% and specificity of 86% if a score ≥ 3 is considered indicative of MIBC). MRI can also be useful to identify macroscopic peri-vesical fat invasion. On the other hand, [^18^F]FDG-PET/CT performance for the detection of tumors in the bladder and the upper urinary tract is hampered by the urinary excretion of [^18^F]FDG. Wang et al. published a meta-analysis regarding the performance of [^18^F]FDG-PET or PET/CT for detecting bladder lesions and reported a sensitivity of 80.0% (95% CI: 71.0 vs. 87.0%) and a specificity of 84.0% (95% CI: 69.0 vs. 93.0%) [10]. To overcome the limitations induced by urinary activity, several studies have proposed adapted protocols to improve [^18^F]FDG-PET/CT’s performances. These include oral hydration with refilling proposed by Higashiyama et al., which may increase the sensitivity to 92% and specificity to 87% [11], and the forced diuresis with furosemide proposed by Nayak et al., which may increase the sensitivity to 96% [12]. Some other authors have evaluated the role of early dynamic imaging in [^18^F]FDG-PET/CT. Yoon et al. compared the sensitivity of early dynamic (10 min after injection), whole-body (60 min after injection), and additional delayed (120 min after injection) PET acquisitions in 52 patients. The sensitivities of early dynamic, whole-body, and additional delayed PET for bladder cancer were 85%, 58%, and 61%, respectively [13]. In clinical routine, forced diuresis (20 to 40 mg of furosemide between 0 and 40 min after [^18^F]FDG injection) with good hydration of the patient is the simplest and most commonly used technique. All in all, [^18^F]FDG-PET/CT has currently very limited use for T-staging of MIBC (Figure 1). 

#### 2.1.2. Regional Nodal Staging

Lymph node involvement is frequent in high-risk MIBC, occurring in approximately 25% of patients with T2 tumors and up to 50% with T3 tumors [14,15]. Whether in CT or MRI, the current recommendation suggests defining pelvic lymph nodes as suspects for metastasis when they are larger than 8 mm in the small axis [16]. This cut-off results in a high false-negative rate [14], as many lymph node metastases are smaller, but also in false-positive cases, as some reactive lymph nodes can be significantly enlarged. Studies on the performance of [^18^F]FDG-PET/CT for N-staging of BC reported sensitivities ranging from 23% to 100%, and specificities ranging from 33% to 100% [17,18,19]. These major discrepancies are probably explained by small sample sizes, differences in population characteristics and acquisition protocols, as well as PET/CT systems and interpretation criteria used. In two recent meta-analyses, there were similar results. Ha et al., in their meta-analysis including 14 studies and 785 patients, found that the pooled sensitivity and specificity of [^18^F]FDG-PET/CT for initial pelvic lymph node staging were, respectively, 0.57 [95% CI 0.49–0.64] and 0.92 [95% CI 0.87–0.95] in a per-patient analysis [20]. Subra et al. found that the pooled sensitivity of [^18^F]FDG-PET/CT for detecting lymph node metastasis was 0.57 (95% CI 0.29–0.80) and the pooled specificity was 0.95 (95% CI 0.91–1.00) [21]. In a prospective study by Girard et al. [^18^F]FDG-PET/CT outperformed CT in a per-area analysis with diagnostic accuracies of 84% and 78%, respectively. On a per-patient analysis, combining the PET and the CT components of the PET/CT, they found that an additional 8% of patients could be correctly classified compared to CT alone [22]. In summary, [^18^F]FDG-PET/CT seems to slightly improve sensitivity for the detection of pelvic lymph node involvement compared to CT, but in a non-selected population this improvement may be considered too low since both modalities have good specificity but relatively poor sensitivity. [^18^F]FDG-PET/CT may be particularly useful in selected patients with enlarged lymph nodes seen on morphological imaging, whether to rule out lymph node involvement and enable curative treatment [23], or to confirm lymph node neoplastic spread and reveal potential additional metastases (Figure 2 and Figure 3).

#### 2.1.3. Distant Metastatic Staging

Distant metastases are not rare in patients with MIBC. The most commonly affected sites are extra-pelvic lymph nodes, bone, lungs, liver and peritoneum [24,25,26]. Accurate initial staging is essential for the patients’ management, notably to prevent overtreatment such as futile major surgery, as stated by the EAU-ESMO 2019 consensus [2]. Several studies have evaluated the role of [^18^F]FDG-PET/CT for the detection of distant metastases, with sensitivities ranging from 54 to 87% and specificities ranging from 90% to 97% in a per-patient analysis [25,26,27,28]. In a recent review by Kim, the pooled sensitivity of [^18^F]FDG PET/CT was superior to that of CT (0.75 vs. 0.43), and with similar specificities (0.95 versus 0.96) [29]. [^18^F]FDG PET/CT could also reveal a second primary cancer that conventional imaging could not [30,31]. The presence of [^18^F]FDG-avid extra-pelvic lesions is also an independent predictor of overall survival in MIBC patients [31,32]. 

Thus, the benefit of [^18^F]FDG PET/CT for M-staging of MIBC is well documented, as is its potential impact on patient’s management. In a recent study by Voskuilen et al. including 771 consecutive patients, [^18^F]FDG PET/CT influenced treatment in 18% of patients. For half of them, it involved a shift from potentially curative to palliative treatment [33]. Some ongoing studies such as the phase II EFFORT-MIBC, investigating patients’ management based on [^18^F]FDG-PET/CT, will provide additional data to consider the implementation of the use of [^18^F]FDG-PET/CT in current guidelines [34]. Currently, the role of [^18^F]FDG PET/CT in the management of MIBC patients is presented slightly differently across current international guidelines. While the European Association of Urology (EAU) states that its exact role continues to be evaluated, the expert consensus guideline from both the European Society of Medical Oncology (ESMO) and the EAU declares that [^18^F]FDG-PET/CT should be systematically included in oligometastatic patients staging when considering radical treatment, in order to prevent over treatment [2,6]. On the other hand, according to American guidelines from the American Urological Association (AUA) and the National Comprehensive Cancer Network (NCCN), [^18^F]FDG-PET/CT should be used only in selected patients to characterize indeterminate findings found on CT [35,36] (Figure 4 and Figure 5).

### 2.2. Follow-Up

#### 2.2.1. Neoadjuvant and Induction Chemotherapy

Cross-sectional imaging is recommended to evaluate treatment efficiency at the end of both NAC and induction chemotherapy [2]. Although response rates are quite high, with 40–50% of complete pathological response for primary tumors on cystectomy specimens, some patients progress under treatment and should not undergo radical surgery [37,38].

A few studies have evaluated the performance of [^18^F]FDG-PET/CT in monitoring the response of BC to preoperative chemotherapy. [^18^F]FDG-PET/CT identified primary tumor response (partial or complete) with a sensitivity of 75% to 83% and specificity of 80% to 90% [39,40], and nodal response with a sensitivity of 93% to 100% and a specificity of 17% to 43% [40,41]. For the detection of overall residual disease, the sensitivity was 75% to 90% and the specificity was 67% to 95% [39,40,42]. Regarding the detection of remaining pelvic LN invasion after preoperative chemotherapy, the sensitivity was 46% to 60% and the specificity was 67% to 71% [31,40]. Moreover, a complete metabolic response on [^18^F]FDG-PET/CT was independently associated with a better outcome in terms of cancer-specific survival after a median follow-up of 40 months: hazard ratio of 3.3, IC95% [1.02–10.65] when SUVmax and adapted Deauville criteria were to evaluate the response, and HR 6.32, IC95% [2.06–19.41] when total lesion glycolytic lesion (TLG) measurement was used [42]. [^18^F]FDG-PET/CT provides quite good performance for response monitoring in patients without nodal involvement, and can therefore help to manage neoadjuvant chemotherapy. Nevertheless, a negative [^18^F]FDG-PET/CT after treatment does not exclude the presence of residual lymph node involvement. Its place in evaluating patients after neoadjuvant chemotherapy needs to be evaluated in a specific and prospective way. To the best of our knowledge, only one study investigated the role of [^18^F]FDG-PET/CT in evaluating nodal response after preoperative immunotherapy (pembrolizumab). The authors reported that baseline PET/CT could help to select patients with MIBC who are best suited for neoadjuvant immunotherapy strategies. A mediocre diagnostic performance for residual nodal involvement before RC and ePLND (38% sensitivity) was also found in this setting [43] (Figure 6).

#### 2.2.2. Chemotherapy in a Metastatic Setting

According to Öztürk et al., [^18^F]FDG PET/CT using EORTC criteria performed better than CT interpretation alone based on RECIST criteria in evaluating the response to first-line chemotherapy (cisplatin and gemcitabine) for metastatic bladder cancer [44]. Regarding early response assessment after only 2 cycles of a combination of methotrexate, vinblastine, doxorubicin, and cisplatin (MVAC) in first-line metastatic chemotherapy, [^18^F]FDG-PET/CT predicted progression-free survival and overall survival [45].

#### 2.2.3. Detecting and Restaging Relapse

Recurrence in MIBC after surgery is common, with a poor prognosis. The overall 5-year recurrence-free survival rate is mediocre, ranging from 58% to 81% [46]. A few studies have evaluated the diagnostic performance of [^18^F]FDG-PET/CT in detecting BC recurrence and reported better performances than CT and MRI, with sensitivities ranging from 87 to 92% and specificities ranging from 83 to 94% [47,48,49]. Alongi et al. reported a significant prognostic value of PET-positivity over progression-free survival and overall survival in this setting [47]. Additionally, it was found that [^18^F]FDG PET/CT lead to a significant change in the management of 40% of 286 patients in a retrospective multi-center study [50]. In a recent meta-analysis including 7 studies by Xue et al., the pooled sensitivity and specificity of PET/CT for the detection of recurrent or residual urinary bladder cancer were 94% and 91%, respectively [51]. Because of the lack of high evidence-level data, and despite its good performance for restaging, [^18^F]FDG-PET/CT is not recommended in these situations by today’s guidelines. 

## 3. Future Directions

### 3.1. Treatment Guided by [^18^F]FDG-PET/CT

[^18^F]FDG-PET/CT may be useful at several time points of patients’ management. Firstly, some teams consider that patients with no spread out of the bladder on [^18^F]FDG-PET/CT might benefit from upfront surgery rather than pre-operative chemotherapy. In a large retrospective study with 711 consecutive patients with MIBC from the Netherlands, Voskuilen et al. reported the impact [^18^F]FDG-PET/CT on patient’s management based on their local standards. Their patients underwent both [^18^F]FDG-PET/CT and contrast-enhanced CT. The clinical stage changed because of [^18^F]FDG-PET/CT in 26% of patients, and the recommended treatment strategy was modified in 18% of patients. False-positive results of [^18^F]FDG-PET/CT were found in 8% of their patients [33]. A similar approach is used in some other European centers, such as the Foch Hospital (Suresnes, France) where most cN0M0 PET patients do not receive NAC. Although there is no literature data on the subject yet, it is possible that [^18^F]FDG-PET/CI is less sensitive for some histological variants of urothelial carcinomas, like plasmacytoid UC. In addition, it should be noted that other factors like Ki-67 index on TURBT specimens or circulating tumor cells could be used to guide the management of patients more precisely. In any case, [^18^F]FDG-PET/CT can be decisive in the management of oligometastatic MIBC, as highlighted by recent EAU-ESMO consensus guidelines [2]. In this field, the prospective phase II study EFFORT-MIBC (NCT04724928) is currently ongoing [52]. In this study, patients with a negative standard work-up are divided into three groups according to [^18^F]FDG-PET/CT results: non-metastatic patients are treated according to the current guidelines, polymetastatic patients are treated with the current standard of care and immunotherapy, and oligometastatic patients with three or fewer metastases seen on [^18^F]FDG-PET/CT are treated according to the current standard of care and additional metastasis-directed therapy [52].

The role of [^18^F]FDG-PET/CT to select the best treatment options in between chemotherapy and immunotherapy in MIBC metastatic patients has not been investigated yet. Nevertheless, it has been shown in other types of cancer that predictive markers of response to immunotherapy can be extracted from [^18^F]FDG-PET/CT images. Especially, a high tumor metabolic volume is usually associated with poor response to immunotherapy, while a high tumor SUVmax has been associated with a better response in some cancer types. Thus, no strategy based on PET is currently validated by prospective studies, but research is underway.

### 3.2. PET/MRI

Hybrid PET-MRI imaging combines functional data of PET with the anatomical high image quality of MRI, thus providing great contrast of soft tissue [53]. Catalano et al. compared the diagnostic performance of PET/CT and PET/MRI in 134 patients with various types of cancer including pelvic malignancies. The authors concluded that PET/MRI could overcome the intrinsic limitation of PET/CT in assessing the local extent of the disease [54]. MRI provides good performances for the detection of the spread to peri-vesical fat (T3) [55], and of the bladder muscle invasion (T2), particularly with the generalization of the VI-RADS score use [9]. Up to date, there were two studies on BC using [^18^F]FDG PET/MRI. Rosenkrantz et al., in a prospective pilot study, compared the diagnostic performance of MRI alone versus [^18^F]FDG PET/MRI. The latter showed higher accuracy (95% vs. 76%) in the detection of metastatic pelvic lymph nodes, providing more accurate staging mainly in the case of equivocal findings of MRI alone [56]. Eulitt et al., in a recent pilot study on 21 patients, found [^18^F]FDG PET/MRI may improve the diagnostic accuracy for the staging of bladder cancer [57]. Salminen et al. evaluated the accuracy of [^11^C]acetate PET/MRI in 15 patients with BC for staging and NAC response monitoring. It showed 100% sensitivity, 69% specificity, and 73% accuracy, despite an overall low sensitivity (approximately 20%) for the detection of nodal metastases [58]. This study is limited by the small size and the heterogeneity of the patients’ sample. Lympho-MRI represents another possible approach in this specific clinical setting since its utility for bladder cancer lymph node staging has been investigated by several studies, in particular using ultra-small superparamagnetic iron oxide (USPIO) nanoparticles as a contrast medium [59]. MRI using USPIO does not rely on morphologic characteristics and has the potential to detect micro-metastasis in normal-sized LNs. A recent meta-analysis performed by Woo et al. included twenty-four studies (2928 patients) evaluating the diagnostic performance of MRI for the detection of LN metastasis in BC and prostate cancer. They found studies that used USPIO (*n* = 4) had higher sensitivity (0.86; 95% CI, 0.62–0.96) than did those not using USPIO (*n* = 17; 0.46; 95% CI, 0.35–0.58) [59].

### 3.3. PET Tracers beyond [^18^F]FDG

#### 3.3.1. Limits of Metabolic Radiotracers

[^18^F]FDG is known to accumulate in the cells of the immune system, with uptake intensities that can be multiplied by 3 to 6 when these cells are activated [60]. This can be responsible for false positives when using [^18^F]FDG-PET/CT for staging malignancies, and BC is no exception. Inguinal lymph nodes usually show moderately increased [^18^F]FDG uptake, but are located outside the lymphatic drainage area of BC (except for BC invading the urethra) and are therefore not an issue. On the contrary, activated pelvic lymph nodes, whether due to TURBT or some other reason, may be responsible for false positive results on [^18^F]FDG-PET/CT.

Other PET “metabolic” tracers such as [^11^C]choline, [^11^C]acetate, and [^11^C]methionine have been studied in the management of BC. These studies with limited samples have not demonstrated sufficient improvement to justify their implementation in everyday practice. Kim et al. performed a literature review and a meta-analysis aiming at assessing the diagnostic accuracy of [^11^C]choline and [^11^C]acetate PET/CT for lymph node staging in patients with BC. They included in their analysis ten studies from 2002 and 2015, for a total of 282 patients, and obtained a pooled sensitivity and specificity of 0.66 and 0.89, respectively. Therefore, they concluded that both tracers have a low sensitivity and a moderate specificity for staging BC [61], but are well suited to study primary bladder tumors thanks to a low urinary excretion. Compared to [^18^F]FDG, amino acid tracers such as [^11^C]methionine accumulate less in activated granulocytes and macrophages, whereas [^11^C]choline is mainly taken up by macrophages [62]. Some authors have suggested that [^11^C]methionine may overcome some of the limitations of [^18^F]FDG-PET for tumor detection and characterization because of its increased uptake in tumors (including low-grade ones) but not in inflammatory lesions, and its delayed urinary excretion [63]. The widespread use of [^11^C]methionine is hampered by the need for an on-site cyclotron, and its performance has not yet been evaluated in a significant number of patients.

#### 3.3.2. FAPI-PET

[^68^Ga]Ga-FAPI-46 is a novel PET ligand that targets the fibroblast activation protein (FAP), which can be overexpressed in tumor-associated fibroblast in the tumor microenvironment. FAP inhibitor (FAPI) showed promising results in several types of tumors. In urothelial carcinoma, FAP expression was demonstrated to correlate with tumor aggressivity. Two small pilot studies were recently conducted to evaluate its performance in urothelial carcinoma and BC. Unterrainer et al., in a study of 15 patients, found a high uptake of [^68^Ga]Ga-FAPI-46 in primary tumor and LN metastases (SUVmax 10.6 (range, 4.7–29.1)). In 26.7% patients there were [^68^Ga]Ga-FAPI-46-positive lesions that were missed on standard routine CT imaging. Additionally, 2 patients who had suspicious pulmonary nodules, as well as pelvic lymph nodes previously rated as suspicious of metastatic spread on CT, were correctly ruled out for malignancy by [^68^Ga]Ga-FAPI-46 PET/CT [64]. Novruzov et al. performed a head-to-head comparison of [^68^Ga]Ga-FAPI-46 and [^18^F]FDG PET/CT in 8 patients with BC. Generally, [^68^Ga]Ga-FAPI-46-PET/CT demonstrated significantly higher uptake compared to [^18^F]FDG PET/CT, with higher mean SUVmax (8.2 vs. 4.6; *p* = 0.01) and higher tumor-to-background ratio (metastasis/blood pool 5.3 with [^68^Ga]Ga-FAPI-46 vs. 1.9 for [^18^F]FDG; *p* = 0.001). Additionally, [^68^Ga]Ga-FAPI-46 detected an additional 30% (*n* = 9) of lesions missed by [^18^F]FDG [65]. Thus, early study results suggest that [^68^Ga]Ga-FAPI-46PET may be useful in the management of BC. However, these need to be confirmed in more patients.

#### 3.3.3. ICI Tracers and Targeted Therapies

Because some patients treated with ICIs have good and durable responses, while others have no benefit at all, predicting response to ICIs is now an important issue in the management of many cancers, including BC. Some PD-1 or PD-L1 radiotracers, such as [^89^Zr]Zr-pembrolizumab, have shown predictive value for the response to ICIs and survival in lung cancer [66]. This predictive value of [^89^Zr]Zr-Pembrolizumab PET imaging was demonstrated despite a limited correlation with PD-1/PDL-1 tumor status assessed by immunostaining. This may be explained by the difference between tumor heterogeneity at the microscopic level (assessed by immunostaining) and at the macroscopic level (assessed by imaging). To our knowledge, the performance of ICI radiotracers in BC has not yet been published, but there is potential interest in this setting.

Several targeted therapies and antibody-drug conjugates are currently being used or developed for the treatment of BC. Nectin-4, the target of enfortumab-vedotin, is overexpressed in 83% of bladder UC [67], with lower expression correlating with lower treatment efficacy. Although nectin-4 is not specific for BC, radiolabelled enfortumab may be of interest for predicting treatment response and BC staging. Similarly, Trop-2, the target of sacituzumab-govitecan, a new antibody-drug conjugate for the treatment of BC, is overexpressed in 83% of BC and could be considered for radiolabelling [68]. Finally, Hoang et al. investigated the ability of the radioimmunoconjugate [^89^Zr]Zr-DFO-Panitumumab to detect epidermal growth factor receptor (EGFR) expression in an orthotopic mouse model of BC, with encouraging results. Indeed, EGFR is often overexpressed in BC [69]. Human epidermal growth factor 2 (HER2) is less commonly overexpressed in BC (about 13% of cases), but could be imaged using the radiolabelled agent [^89^Zr]Zr-trastuzumab [70].

### 3.4. Artificial Intelligence

In medical imaging, Artificial intelligence (AI) can be used in image processing, medical database retrieval, artificial neural networks (ANN) to classify images automatically, and computer vision. Moreover, AI can improve the accuracy of abnormality detection, dosage calculations, and the interpretation of findings [71]. AI can be helpful in pathology prediction, detection, the prediction of early metastatic disease, survival estimation, and post-therapeutic evaluation [72]. Furthermore, AI can enhance [^18^F]FDG PET imaging quality by improving the attenuated correction even with structural imaging [72]. However, AI in PET has its limitations. First, AI requires massive interpreted data for development and learning, making it less reliable for small datasets. Second, the reliability and reproducibility of AI algorithms are needed. As modelling continues to be complicated, the AI’s ‘black box’ nature makes the results of various AI models challenging to comprehend and explain. At the moment, trusted healthcare AI prefers the stability and explicability of unknown and diverse data [73]. Generally, despite the disadvantages mentioned above, AI is important in PET, including the administration and synthesis of drugs, the management of patient information, the interpretation of reports, and image processing and acquisition. Moreover, AI can help researchers carry out investigations and identify novel molecular biomarkers [74]. In a recent study, a machine learning algorithm based on manually measured PET and CT features performed as well as physicians in detecting pelvic lymph node involvement of LN on [^18^F]FDG PET/CT at initial staging for MIBC [75]. To date, there is no published study on the use of fully automated deep learning methods on [^18^F]FDG PET images for BC, but promising results of its application using CT and MRI images have been reported in terms of predicting the depth of invasion of the primary tumor [76], grade [77], local and systemic staging [78], and the assessment of treatment response [79]. Additionally, AI based on PET/CT images has been used in other malignancies to predict nodal disease [80], risk stratification [81], treatment response [82], and patient outcomes [83]. 

## 4. Conclusions

The management of metastatic MIBC has evolved significantly in recent years with the advent of immunotherapy and targeted therapies, whereas the management of patients with non-metastatic MIBC still relies on neoadjuvant chemotherapy in those who are suitable to receive it. Prognostic and predictive data are needed more than ever in this context, both from imaging and non-imaging techniques, to better personalize patient management. 

Imaging provides important information in the staging and restaging of patients with BC. CT is the most frequently used imaging modality for BC. MRI offers additive value for the evaluation of muscular invasion and the local staging, which is critically important to optimize patient’s management. The main interest of [^18^F]FDG-PET/CT lies in its ability to detect distant metastases. At the lymph node level, its performance is slightly better than that of contrast-enhanced CT. However, it is not yet established how to use these gains in clinical routine. Prospective studies are underway and may provide information on this subject in the near future, as well as on [^18^F]FDG-PET/CT usefulness in restaging and therapeutic evaluation of BC patients. 

Other novel imaging techniques in BC management include bladder MRI, mainly for primary tumor assessment, and [^68^Ga]Ga-FAPI-46I-PET, which has shown good performance in detecting distant and nodal metastases (Figure 7). Radiolabelled antibody-drug conjugates, ICIs or targeted therapies may also be useful in the future of BC management.

## Figures and Tables

**Figure 1 pharmaceuticals-16-00606-f001:**
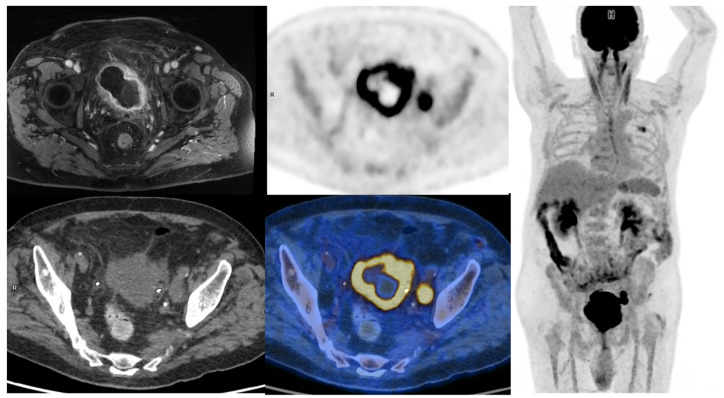
65-year-old patient with a locally advanced MIBC. Up and left: T1 fat-sat after gadolinium contrast media injection MRI sequence, showing an abnormal enhancement of peri-vesical fat (local stage T3b). Other: PET, CT, and PET/CT fused images revealed a necrotic bladder mass associated with a supra-centimetric ilio-obturator adenopathy on the left side, both with intense FDG uptake. No extra-pelvic lesion was seen (MIP showing focal uptake in the left lung of infectious origin). The patient was managed with first-line chemotherapy, followed by surgery after a good response to treatment.

**Figure 2 pharmaceuticals-16-00606-f002:**
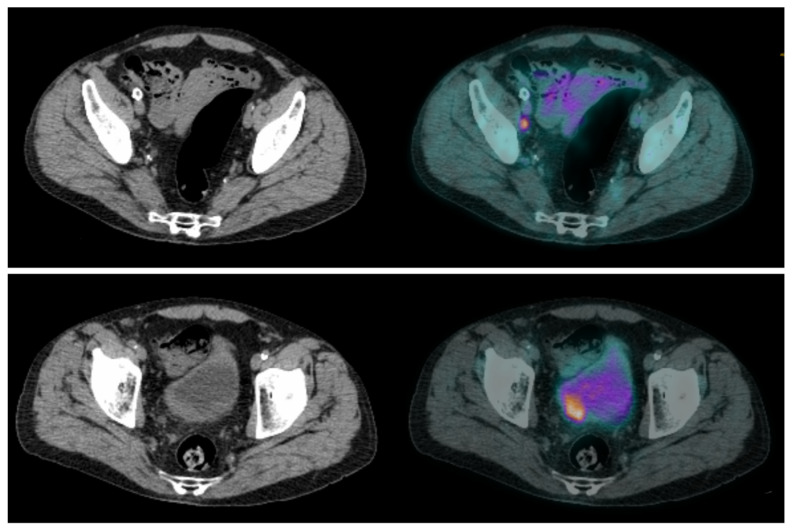
CT and fused FDG-PET/CT images performed at initial staging of a MIBC for a 64-year-old patient. (**Bottom line**): Intense uptake of the residual primary tumor that was found pT3a at pathological analysis of the cysto-prostatectomy specimen. (**Top line**): Intense uptake of a single external iliac lymph node of 7 mm of short axis, which did not match criteria to be considered cN+ on CT alone. This lymph node was found pN+ at pathological analysis of the dissection specimen.

**Figure 3 pharmaceuticals-16-00606-f003:**
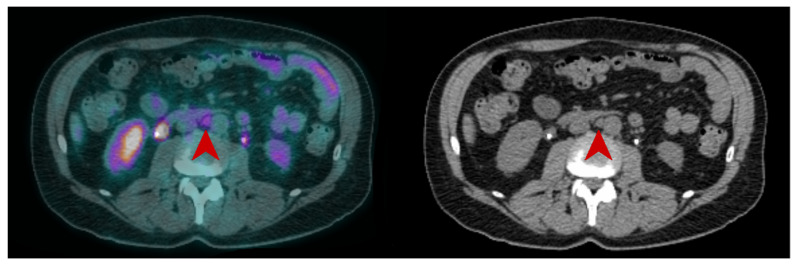
CT and fused FDG-PET/CT images performed at initial staging of a MIBC for a 62year-old man. The patient was classified cN+M0 according to CT alone, and FDG-PET/CT images revealed a moderate uptake of an infra-centimetric lumbar adenopathy (red arrow) that involved restaging to cM1a.

**Figure 4 pharmaceuticals-16-00606-f004:**
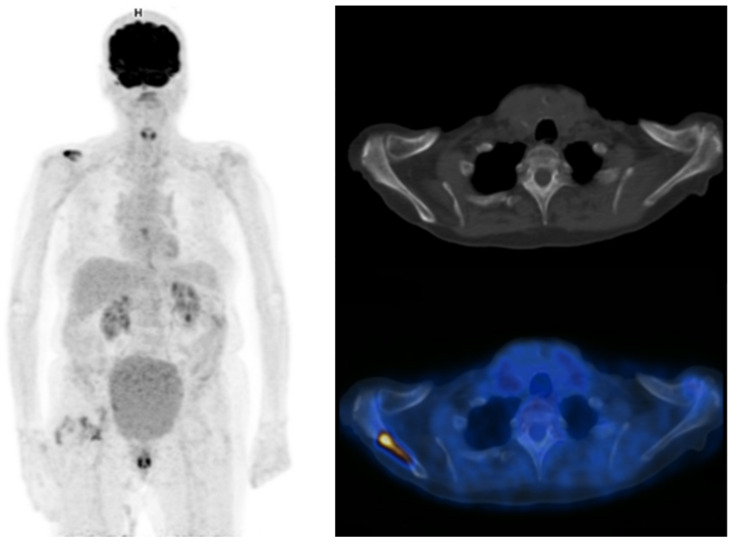
Maximum intensity projection PET, CT and fused FDG-PET/CT images performed at initial staging of a MIBC for an 87-year-old man. The patient was initially classified cN0M0 according to CT alone, and was then upstaged M1b with FDG-PET/CT that highlighted an unknown bone metastasis.

**Figure 5 pharmaceuticals-16-00606-f005:**
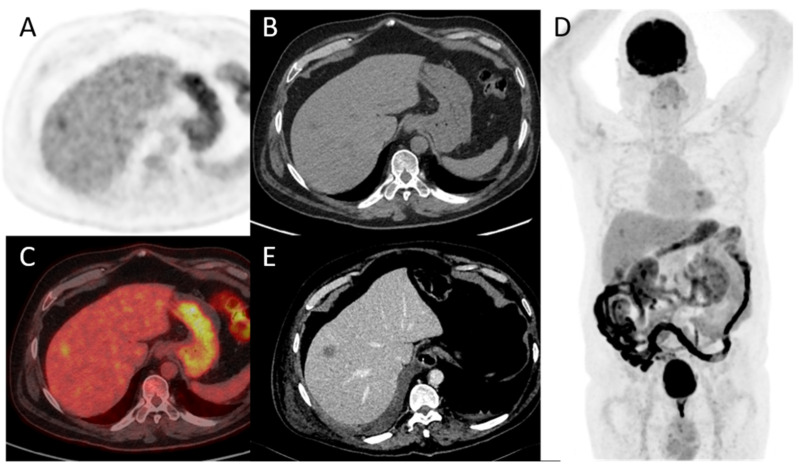
A 78-year-old patient with a diagnosis of urothelial carcinoma. (**A**–**D**) PET, CT, fused PET/CT, and PET maximum intensity projection of baseline FDG-PET/CT, respectively. A discreet liver focal liver uptake corresponds to a small hypodensity in unenhanced CT. It was overlooked at the time of diagnosis and the patient underwent a cystectomy with an enterocystoplasty procedure. (**E**) Follow-up contrast-enhanced CT at 3 months shows an enlargement of the hepatic hypodensity, evocative of its metastatic origin. This case shows that although liver metastases are rare at the time of bladder cancer diagnosis, their detection is paramount.

**Figure 6 pharmaceuticals-16-00606-f006:**
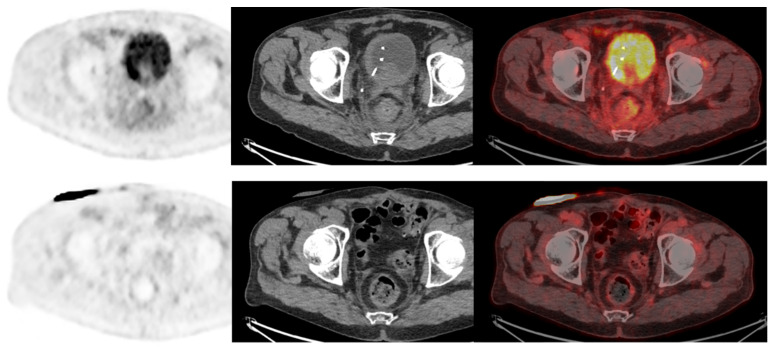
A 73-year-old patient with a diagnosis of urothelial carcinoma. (**Upper row**): Baseline FDG-PET/CT shows a diffuse pararectal et mesorectal infiltration, with a faint FDG uptake, surgically confirmed as peritoneal metastasis. (**Lower row**): Same patient seen after 6 cycles of neoadjuvant chemotherapy. FDG-PET/CT shows a complete regression of the posterior pelvis infiltration. The case illustrates the infiltrative nature of low tumor cell density in rare urothelial carcinoma cases, requiring special attention when FDG-PET/CT is used as a systemic staging modality.

**Figure 7 pharmaceuticals-16-00606-f007:**
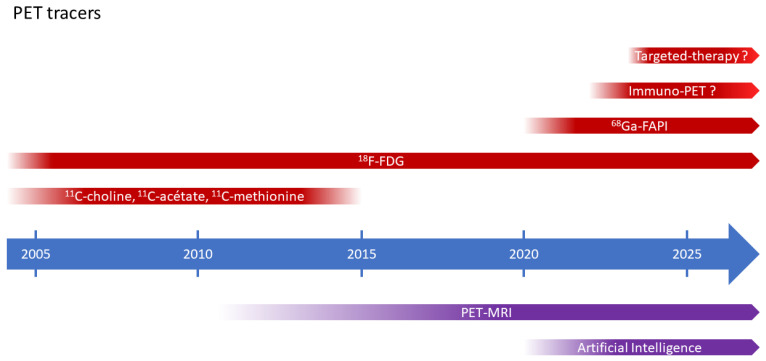
Schematic cartoon showing different PET tracers.

## Data Availability

Not applicable.

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
