# Peer review of "PET Imaging in Bladder Cancer: An Update and Future Direction"

_pharmaceuticals, 2023, doi:10.3390/ph16040606_

Round 1

Reviewer 1 Report

This manuscript, submitted by Jules ZHANG-YIN et al., deals with " PET imaging in bladder cancer, an update, and future direction ". This article mainly focuses on the role of 18F-fluorodeoxyglucose positron emission tomography in the clinical management of bladder cancer patients, especially for staging and follow-up; treatment guided by FDG PET/CT; the role of FDG PET/MRI, the other PET radiopharmaceuticals beyond FDG, such as 68Ga- or 18F-labeled fibroblast activation protein inhibitor; and the application of artificial intelligence.

The review paper under evaluation is well-written, and comprehensive explanations were detailed. Therefore, I suggested the acceptance after minor corrections.

1.    Text font irregularity in the introduction of the manuscript.

2.    Adding a few more figures to the MS might improve the manuscript.

3.    I suggest adding the literature time-period for preparing the review. Also, a brief description of how the authors selected information from the literature in the databases.

Author Response

Thank you for your comments which are all valuable and very helpful for revising and improving our paper.

We have studied comments carefully and have made correction which we hope meet with approval.

  1. Text font irregularity in the introduction of the manuscript

R: This has been corrected (Times New Roman, 12)

  1. Adding a few more figures to the MS might improve the manuscript.

R: We have added 3 more figures. Two clinical cases (Fig 5 and 6) and a schematic cartoon showing different PET tracers both previous and future (Fig 7).

  1. I suggest adding the literature time-period for preparing the review. Also, a brief description of how the authors selected information from the literature in the databases.

R: Thank you for this highly relevant suggestion. The literature covers essentially papers published after 2010 and there were only few “major papers” before this date that we selected: such as Kibel AS et al. JCO 2009 or Kaufman et al. Lancet 2009. This is a narrative review, so there were no precise criteria for selection of data in the literature. All the authors have good clinical experiences in the field, the selection is based on our knowledge.  

We have added a sentence at the end of the introduction: “We have searched in the database of MEDLINE using the key words of “PET” and “bladder cancer”. The literature covers essentially papers published after 2010. All authors participated, thanks to their experience in the field, of selecting the papers."

Reviewer 2 Report

In this review article by YIN et al, the authors have analyzed the literature for FDG based PET-positron emission tomography for management of bladder cancer, especially with respect to clinical staging, follow-up and treatment based on FDG-PET-CT. Though the authors have given a comprehensive idea about PET, authors can include some minor points, which would improve the manuscript. 

In the article, the authors could include metabolic prognostic factors in the management of bladder cancer.

Schematic cartoon showing different PET tracers both previous and future could be better in the review. 

The authors could also comment on other methods of diagnosis in bladder cancer in their conclusion.

Author Response

Thank you for your comments which are all valuable and very helpful for revising and improving our paper.

We have studied comments carefully and have made correction which we hope meet with approval.

- In the article, the authors could include metabolic prognostic factors in the management of bladder cancer.

R: Thank you for this suggestion. We added a sentence in the section II.2.3 "Alongi, et al. reported a significant prognostic value of PET-positivity over progression-free survival and overall survival in this setting [47]."

- Schematic cartoon showing different PET tracers both previous and future could be better in the review. 

R: It is a highly relevant suggestion. We added this schematic cartoon (figure 7).

- The authors could also comment on other methods of diagnosis in bladder cancer in their conclusion.

R: We added a short paragraph on CT and MRI with highlights of their specificities. “Imaging provides important information in the staging and restaging of patients with BC. CT is the most frequently used imaging modality for BC. MRI offers additive value for the evaluation of muscular invasion and the local staging, which is critically important to optimize patient’s management.”

Reviewer 3 Report

The article is completely out of the scope both the Pharmaceuticals journal and the Special Issue “Application of Radiolabeled Agents: Imaging, Therapy, Multimodal Approaches and Beyond”. The Pharmaceuticals journal deals with any aspects of Drug Discovery and Drugs approval, however the reviewed article is 100% clinical devoted to review of PET with well known and approved radopharmaceuticals for diagnostic of bladder cancer. The paper is well written and clinically valuable but completely out of the scope of the particular submitted journal and Issue. I’d like to recommend and encourage the authors to submit their manuscript to some other CLINICAL journal.

Author Response

Thank you for your commentary and suggestion. Indeed, this is a 100% clinical devoted review. The editor will make the final decision on choice of section.